

# Understanding colloidal behavior in açaí (*Euterpe oleraceae*)-based nanoemulsions

Mikaela Ferreira[1,2], Leandro Machado Rocha[3], Rodrigo Cruz[1], Francisco Paiva Machado[3], Celia Machado Ronconi[4] and Caio Fernandes[1,2,3]

[1] Laboratory of Phytopharmaceutical Nanobiotechnology, Federal University of Amapá, Macapá, Amapá, Brazil
[2] Department of Biological and Health Sciences, Federal University of Amapá, Macapá, Amapá, Brazil
[3] Laboratory of Technology in Natural Products, Fluminense Federal University, Niterói, Rio de Janeiro, Brazil
[4] Chemistry Institute, Universidade Federal Fluminense, Niterói, Rio de Janeiro, Brazil

Corresponding author
Caio Fernandes, caiofernandes@unifap.br

## ABSTRACT

**Background.** In recent years there has been growing interest in novel, eco-friendly, natural product-based colloids, including nutraceuticals and cosmetics. Despite the Amazon region's globally recognized biodiversity, efforts to sustainably use natural products for viable applications remain at the forefront of innovation. The superfruit *Euterpe oleracea* Mart, commonly known as açaí, is recognized by its high level of phenolic compounds and valuable oil. However, to the best of our knowledge, studies on its colloidal chemistry and the combination of its oil with phenolic-rich extract are scarce. Therefore, the aim of this study was to evaluate the generation of nanoemulsions containing açaí oil and phytoglycerol extract through a low-energy method, investigating the influence of surfactant's nature and ratio on the droplet formation and stabilization, enhancing the potential for sustainable product development.

**Methods.** Total phenolics in the açaí-based phytoglycerol extract obtained from Heide Extratos Vegetais® were quantified using the Folin-Ciocalteau/Basf method. The nanoemulsions containing açaí oil and phytoglycerol extract were evaluated using individual or blended non-ionic surfactants with hydrophilic-lipophilic balance (HLB) values ranging from 9 to 15. The nanoemulsions consisted of 90% (w/w) aqueous phase (5% phytoglycerol extract in deionized water) and 10% (w/w) oily phase (varying ratios of açaí oil to surfactants, 7:3 to 9:1). The formulations were monitored for color, appearance, and p signs of instability. Additional systems were then prepared using twice the amount of oily phase, followed by a 1:1 dilution in the aqueous phase to achieve the same concentrations as previously reported. The optimal systems were subjected to dynamic light scattering analysis at room temperature (25 °C) and after a linear temperature ramp.

**Results.** The spectrophotometric characterization revealed that the phytoglycerol extract contains $2.8 \pm 0.4$ mg/mL of gallic acid equivalents. Various instability phenomena were observed at most surfactant-to-oil ratio (SOR) and with different surfactants. However, suitable açaí-based nanoemulsions were achieved using polysorbate 85 (HLB 11) and a combination of polysorbate 80/sorbitan monooleate (HLB 13) at an SOR of 9:1. The nanoemulsions with this pair of surfactants exhibited smaller droplet sizes (*ca* 200 nm) and showed no statistically significant difference in size during storage. The slope of size versus temperature also indicated that the nanoemulsion with polysorbate 80/sorbitan monooleate (HLB 13) at an SOR of 9:1 was more stable that the system prepared solely with polysorbate 85 (HLB 11).

## INTRODUCTION

In the last two decades, the beauty and cosmetics market has shown global growth (*Petruzzi, 2023*), with personal care products being the largest segment in 2022 (*Killip, 2023*). Skincare products stood out, contributing more than 38% to the global cosmetic market in 2022 (*Grand View Research, 2023*).

Environmental issues have gained significant attention in recent years, leading to the growth of the green movement, which promotes a sustainable and harmonious lifestyle with the planet, valuing natural products (*Chin et al., 2018*). With increasing interest in skincare cosmetics and environmental awareness, consumers are attentive to formulations, often opting for less synthetic and cruelty-free products. This shift is due to the highlighted negative effects of synthetic materials on the health and the environment, prompting improvements in formulations that prefer vegetable derivatives whenever possible (*Iha et al., 2008*; *Amberg & Fogarassy, 2019*).

This trend has quickly influenced cosmetology, where natural products are increasingly used in formulations for their sustainability and pharmacological properties. Major cosmetic companies are transforming their portfolios by exploring biotechnology and raw materials, and adopting sustainable business models to achieve goals such as carbon neutrality (*L'Oreal, 2023*). Thus, innovation is the cosmetic industry worldwide, particularly with natural products, is imperative.

Açaí (*Euterpe oleracea* Mart) is a palm tree from Amazon rainforest of great importance to Brazil's northern region, where its fruits are widely consumed due to their high nutritional value. The beneficial effects of açaí have led to its popularity beyond the Amazon, spreading to other Brazilian states and internationally, resulting in the export of several açai-based products (*Galotta & Boaventura, 2005*; *Ferrari & da Rocha-Filho, 2011*).

Several studies have demonstrated the diverse pharmacological activities of E. *oleracea*, varying depending on the plant part used and its chemical composition (*de Almeida Magalhães et al., 2020a*). Notable activities include antioxidant (*Pacheco-Palencia, Mertens-Talcott & Talcott, 2008*; *Pacheco-Palencia, Duncan & Talcott, 2009*; *Kang et al., 2011*; *Sotero et al., 2013*; *Petruk et al., 2017*; *Teixeira-Costa et al., 2020*), antimicrobial (*de Almeida Magalhães et al., 2020b*), and anti-inflammatory (*Favacho et al., 2011*; *Kang et al., 2011*), related to both the extract and açaí oil. This interest is due to the high quantity and variety of phenolic compounds in E. *oleracea* fruits, such as anthocyanins (*Xiong et al., 2020*; *Costa et al., 2021*; *de Oliveira et al., 2021*) and flavonoids (*Kang et al., 2011*), and long-chain unsaturated fatty acids in its oil, such as oleic, palmitic and linoleic acids (*de Almeida Magalhães et al., 2020b*; *Matta et al., 2020*). Phenolic acids like procyanidin dimers and trimers, vanillic, syringic and p-hydroxybenzoic acids are also found in E. *oleracea* fruit oil (*Pacheco-Palencia, Mertens-Talcott & Talcott, 2008*). These characteristics make açaí highly interesting for phytocosmetic formulations.

Nanotechnology, an emerging research field, involves the development, characterization, and application of materials on the nanometric scale and has diverse applications (*Shahcheraghi et al., 2022*). Recent years have seen extensive scientific studies on various nanostructured systems, including nanoemulsions, due to their wide applicability in the cosmetics industry and potential for innovative product development (*Gupta et al., 2022*).

Nanotechnology offers several advantages, such as encapsulating and protecting bioactive substances, modifying optical, rheological, and stability properties due to reduced droplet size and greater surface area, and reducing toxicity, thus improving product functionality (*Jafari & McClements, 2018*; *Torres M et al., 2019*). It is also a great alternative for incorporating natural products, allowing the delivery of hydrophobic bioactive substances in an aqueous medium, with low-cost and easily reproducible preparation methods, making nanostructured systems suitable for industrial scale.

This work focuses on nanoemulsions, heterogeneous systems with a translucent or transparent appearance, constituted of oily and aqueous phases, where one is uniformly dispersed in the other as spherical droplets with a radius of less than 100 nm. Conventional emulsions are commonly used in cosmetic products. However, advances in nanotechnology research have made its application in cosmetic essential, allowing the development of more effective formulations with unique sensory properties (*Daudt et al., 2013*). Over the years, several companies have released nanocosmetics, with Lancôme, a luxury subdivision of L'Oreal, pioneering the field by releasing a cream containing vitamin E nanocapsules in 1995 (*Baril et al., 2012*).

The benefits of using nanotechnology in cosmetics are linked to the reduced dimensions of the systems. Nanoemulsion droplets and the surfactants can interact with intracellular structures, increasing permeation in the stratum corneum, enhancing product efficiency (*Santos et al., 2019*), and providing greater retention time of active components on the skin, enhancing cosmetic effects.

A main gap for complex mixtures of natural products in cosmetics is the impaired stability when compared to single synthetic compounds. In this context, herbal nanoformulations for these purposes open perspective for better performance due to efficient penetration, sustainable release of bioactive compounds, greater stability, enhanced efficacy and less toxicity and quantity, the last also being in accordance with sustainable concepts in which less amounts of bioactive compound with no impairment of bioactivity would benefit the industry in accordance with standing tree (*Vaishampayan & Rane, 2022*).

A nanoemulsion prepared with açaí oil was prepared, and only at a high surfactant-to-oil ratio (9:2) was possible to reach size around 117 nm (estimated by dynamic light scattering (DLS)). In this study, phase inversion temperature method was used and solely polysorbate 80 was chosen as surfactant (*Monge-Fuentes et al., 2017*). Another study aiming to generate açaí nanoemulsions showed that Ceteareth-20 at high concentrations (7–10%) by phase inversion composition (PIC) method was capable of generating nanodroplets around 150–200 nm (*Loureiro Contente et al., 2020*). It is worth mentioning that heating impairs the costs of the process and also may be limiting due to possible degradation of natural products, such those present in açaí oil.

Given the changing consumer profile and the potential of nanotechnology for innovation in cosmetics, this work focuses on developing açaí-based nanoemulsions for potential cosmetic use. It addresses the need for high-value products that value national flora and sustainability.

## MATERIALS & METHODS

### Materials

The non-ionic surfactants (polysorbate 80, polysorbate 85, sorbitan monooleate and sorbitan trioleate) and gallic acid were obtained from Sigma-Aldrich (St. Louis, EUA). Organic solvents were obtained from Vetec (Rio de Janeiro, Rio de Janeiro). Phytoglycerol extract and oil from açaí fruit were kindly donated, by Heide Extratos Vegetais (Pinhais, Paraná) and Luna Green Bioativos (Goiânia, Goiás), respectively.

### Quantification of phenolics

The phenolic compounds in the phytoglycerol extract were quantified using the Folin-Ciocalteu (FCR/Basf) method (*Chen, Cheng & Liang, 2015*). Prior to the assay, a 1:5 aqueous solution of the phytoglycerol extract (S1), a one mol/L solution of the Folin-Ciocalteu reagent (S2), and a 20% (w/v) sodium carbonate ($Na_2CO_3$) aqueous solution (S3) were prepared. Aliquots of S1 (250 µL) and S2 (250 µL) were sequentially added in a five mL volumetric flask. After 5 min, an aliquot of S3 (500 µl) was added, and the volume was completed with deionized water. After 25 min, the analysis was carried out using a UV-Vis spectrophotometer (2600 UV-Vis, Shimadzu, Kyoto, Japan) at 730 nm. The same procedure was performed for a blank, replacing the phytoglycerol extract by glycerol, and for gallic acid (GA). A standard curve was constructed with this phenolic acid and the results were expressed as GA equivalent (mg GAE/mL) as follows:

$$C = \frac{(Abs - b) \times F_D}{a}$$

where $C$ represents phenolic compounds, $Abs$ is the absorbance of the sample, $a$ and $b$ are the angular and linear coefficients, respectively, and $F_D$ is the dilution factor.

### Nanoemulsions

The açaí-based nanoemulsions were prepared by a low-energy method. The technique involved the dropwise addition of the aqueous phase to the oily phase under vortex stirring. Final mass was two g and the aqueous phase corresponded to 90%, (w/w) of the system, being constituted by 1.71 g of deionized water (95%, v/v) and 0.09 g of phytoglycerol extract (5%, v/v). Oily phase corresponded to 10% (w/w) of the system and was constituted by non-ionic surfactant (s) and açaí oil at different surfactant-to-oil ratios. The presence of bluish reflections was considered an indicator of nanostructure formation. Intrinsic factors related to the physical incompatibility of the components were evaluated, such flocculation, creaming, sedimentation and/or phase separation.

## Factor of influence
### Surfactants
The nanoemulsions were prepared using different non-ionic surfactants (individually or in pairs): polysorbate 80, polysorbate 85, polysorbate 80 + sorbitan monooleate, and polysorbate 80 + sorbitan trioleate. The hydrophilic lipophilic balance (HLB) values of the mixtures ranged from 9 to 15. The equation below was used to determine the required mass of each surfactant in the mixtures for a certain HLB:

$$HLB_R = \frac{HLB_A \times m_A + HLB_B \times m_b}{m_f}$$

where $m_f$ represents the final mass of the surfactant, which varied according to the surfactant:oil ratio (SOR); $m_A$ and $m_B$ represent the mass of the surfactants; $HLB_R$ is the desired HLB; $HLB_A$ and $HLB_B$ represent the HLB values of the utilized surfactants.

### Surfactant:oil ratio
The influence of different ratios of surfactant (s) and açaí oil in the oily phase was evaluated using the following surfactant-to-oil ratio: 7:3, 8:2 and 9:1. These ratios corresponded respectively to 7, 8 and 9% (w/w) of surfactant (s), and 3, 2 and 1% (w/w) of açaí oil in the nanoemulsions.

### Prototypes preparation
After the development procedure, the nanoemulsions with the best macroscopic characteristics were selected for the prototype preparation for future cosmetic application. This involved a modified nanoemulsion formulation containing the double of the oily phase while keeping the final mass at two g. After its preparation, the concentrated nanoemulsions were diluted 1:1 proportion with the aqueous phase (distilled water + 5% phytoglycerol extract) and analyzed on the day of preparation (day 0), day 1 and day 7 after storage at room temperature.

### Dynamic light scattering analysis
The prototypes were characterized by dynamic light scattering (DLS) using a ZetaSizer Advance Lab Blue (Malvern Instruments®, Worcestershire, UK) equipment for droplet size (d/nm), zeta potential (mV) and polydispersity index (PdI) measurements. The samples were diluted in deionized water to avoid multiple light scattering, and all analyses were performed in triplicate. Equipment settings were as follows: detector, He-Ne (633 nm); measurement angle, 90 °C; temperature, 25 °C with return to default temperature; equilibration time, 120 s; data processing, general purpose with accordance to the majority of dispersions and emulsions, being used with all unknown samples. Additionally, the nanoemulsions were subjected to a linear temperature ramp from 25° to 65 °C, in 5 °C intervals (*Ortiz-Zamora et al., 2020*). Based on the results, the droplet size variation at the end of the temperature i was calculated using the equation below, where $V$ is the droplet variation; $T_i$ and $T_f$ represents the initial and final temperatures, respectively:

$$V = 100 \times \frac{[mean\ droplet\ size\ (T_i) - mean\ droplet\ size\ (T_f)]}{mean\ droplet\ size\ (T_f)}.$$

### Statistical analysis

One-way analysis of variance (ANOVA) followed by the Tukey test was conducted in GraphPad Prism v.8 to analyze droplet mean size and PdI variation during the thermal stress and over time.

## RESULTS

In order to determine the total phenolic concentration in the phytoglycerol extract, the Folin-Ciocalteu method (*Folin & Denis, 1915*), known for its reproducibility and low cost, was used with minor modifications (*Chen, Cheng & Liang, 2015*). The phytoglycerol extract used in the study was constituted by $2.8 \pm 0.4$ mg/mL content, expressed as gallic acid equivalent. Some phenolics already described in açaí are glycosides (*Petruk et al., 2017*), which may contribute to the generation/stabilization of the colloid due to their polar moiety coupled with a portion that is intrinsically low-intermediate water-soluble (*Velderrain-Rodríguez et al., 2021*). Additionally, studies have demonstrated that modifying the aqueous phase is a suitable strategy for obtaining nanoemulsions. Glycerol (0–50%) aqueous solution can nanoemulsify vitamin E medium triglyceride chains (8:2) with monomodal distribution. This polyol can alter physicochemical and molecular parameters of the dispersed system, therefore, influencing the droplet size (*Saberi, Fang & McClements, 2013b*). In the present study, the phytoglycerol extract was used for the nanoemulsification of açai oil.

Different HLB values (nine to 15) were achieved through the combination of different surfactants, allowing the influence of the surfactant type to be verified. The surfactant-to-oil ratios (SOR) ranged from 7:3 to 9:1. In the nanoemulsions prepared with a 7:3 SOR, a milky, opaque appearance and yellow color were predominantly observed for the entire range of established HLBs. Macroscopic analysis highlighted their susceptibility to instability phenomena, especially flocculation and creaming. In some cases, such as the nanoemulsions at HLB 11 prepared with polysorbate 85, phase separation was observed few minutes after preparation. The exception was the nanoemulsion at HLB 13 prepared with polysorbate 80 and sorbitan monooleate, which was less opaque and more fluid, with theses aspects remaining stable during the analyses.

Similar behavior was observed for the nanoemulsions obtained at 8:2 SOR. For both s of surfactants, the nanoemulsions at HLB 9, 10, 14 and 15 rapidly showed signs of instability, such as creaming and color change, which became more pronounced throughout the analysis period. Flocculation was observed at all HLBs for the polysorbate 80 + sorbitan trioleate pair. Regarding the nanoemulsion at HLB 11 with, polysorbate 85, the main difference was the presence of flocculation and creaming instead of phase separation. The nanoemulsion at HLB 13 with polysorbate 80 + sorbitan monooleate presented overall better characteristic than the aforementioned formulations, exhibiting fluidity and a more translucent appearance.

A study carried out with sucupira-branca oil revealed that polysorbate 80+sorbitan trioleate revealed the ability of this surfactant to inhibit Ostwald ripening and inducing kinetic stability (*Ortiz-Zamora et al., 2020*). Differing from this previous study, the açaí oil

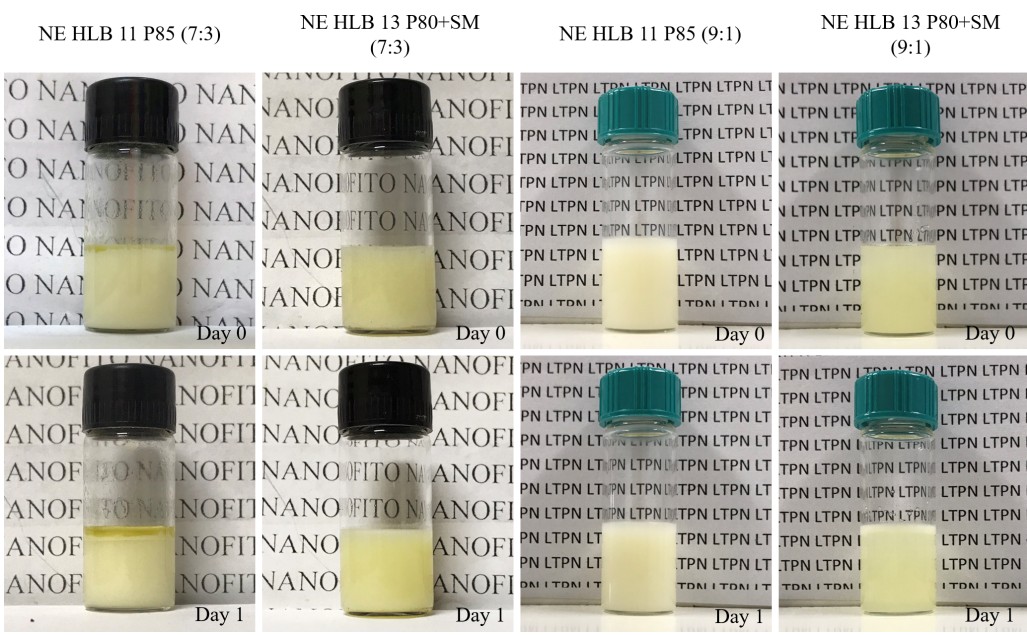

**Figure 1** Systems prepared at SOR 7:3 with polysorbate 85 (HLB 11) and polysorbate 80 + sorbitan monooleate (HLB 13).

is rich in phenolic compounds that can be considered more hydrophilic than conventional components of natural oils. According to *Cottrell & van (2014)*, sorbitan trioleate has a HLB value of 1.8, while sorbitan monooleate has a HLB value of 4.3. Therefore, considering the classical Griffin scale, the higher polarity of sorbitan monooleate is probably the reason for better compatibility with açaí oil in the present study, being both together with polysorbate 80 considered food grade constituents that therefore are suitable for preparation of a prototype of nanocosmeceutical.

Figure 1 exemplifies some macroscopical behavior of the açaí nanoemulsions prepared at the present study. Polysorbate 85 is a non-ionic surfactant of intermediate HLB value. In the present study, at lower SOR values the destabilization may be due to main effects when compared to the stable generated system at SOR 9:1 prepared with polysorbate 80+sorbitan monooleate at HLB 13. First, the amount of surfactant was insufficient at lower SOR to stabilize the nanoemulsions, since this ingredient acts by reducing the interfacial tension and also generating a thin layer capable of providing a steric repulsion and adsorbing ions capable of modulating zeta potential. Lastly, the presence of higher amounts of surfactant blend with a suitable amount of sorbitan monooleate may strengthen the film, avoiding Ostwald ripening, that would be expected for a more hydrophilic oil.

At SOR 9:1, the systems obtained at lower HLBs (9 and 10) presented flocculation, creaming and intense opacity. At higher HLBs (14 and 15), the gradual addition of the aqueous phase led to the generation of highly viscous systems, impairing homogenization. Although the Tyndall effect was visible against the light, rapid destabilization was observed few minutes after preparation. It is well known that some transitory high-viscous systems

**Table 1  Dynamic light scattering analysis of the prototypes of açaí-based nanoemulsions.**

| Day | Mean hydrodynamic size (nm) | | Polydispersity index | |
|---|---|---|---|---|
| | HLB 11 | HLB 13 | HLB 11 | HLB 13 |
| 0 | $244.3 \pm (6.958)$[a] | $216.9 \pm (0.7893)$[a] | $0.181 \pm (0.0222)$[a] | $0.246 \pm (0.0204)$[a] |
| 1 | $275.6 \pm (8.055)$[b] | $209.1 \pm (6.351)$[a] | $0.203 \pm (0.0838)$[a] | $0.261 \pm (0.0280)$[a] |
| 7 | $283.8 \pm (6.255)$[b] | $208.3 \pm (3.163)$[a] | $0.251 \pm (0.028)$[a] | $0.203 \pm (0.0065)$[b] |

**Notes.**

Different letters in the same column represent a significant difference between the data ($p < 0.05$). *HLB 11 size: 0 *versus* 1 $p$ value = 0.0041; 0 *versus* 7 $p$ value = 0.0012; 1 *versus* 7 $p$ value = 0.3946. HLB 11 PdI: 0 *versus* $p$ value = 0.8683; 0 *versus* 7 $p$ value = 0.3048; 1 *versus* 7 $p$ value = 0.5388. HLB 13 size: 0 *versus* 1 $p$ value = 0.1288; 0 *versus* 7 $p$ value = 0.951; 1 *versus* 7 $p$ value = 0.9695. HLB 13 PdI: 0 *versus* 1 $p$ value = 0.6584; 0 *versus* 7 $p$ value = 0.0913; 1 *versus* 7 $p$ value = 0.0300.

may play a crucial role in the low energy nanoemulsification methods. One would expect that if the oil is not promptly and homogeneously incorporated, higher polydispersity will be observed (*Feng et al., 2020*).

At HLB 12 for both pairs, and also at (i) HLB 11 prepared solely with polysorbate 85 and (ii) at HLB 13 prepared with polysorbate 80 + sorbitan monooleate, satisfactory characteristics were observed. Two main mechanisms are important for the generation and stabilization of a nanoemulsion through the utilization of surfactants. The first is related to the reduction of interfacial tension, which facilitates generation by enhancing SOR. The second is related to the ability of the surfactant to generate a resistant film.

*Pourshamohammad et al. (2024)* showed that low SOR ($<1$) only generate high size ($>300$ nm) and high PdI for nanoemulsions prepared with *Carum copticum* essential oil at a 10:90 oily phase to water phase ratio, which was similar to the used in the present study, even in the presence of various concentrations of ripening inhibitors.

Therefore, we selected nanoemulsions at SOR 9:1 prepared with polysorbate 85 (HLB 11) and polysorbate 80 + sorbitan monooleate (HLB 13) as prototypes of açaí-based nanoemulsions with single or pair of surfactants. The preparation of a concentrated system with a further dilution step, as performed in this study may be considered a suitable strategy for enhancing kinetic stability (*Saberi, Fang & McClements, 2013a*). Moreover, aiming for a further industrial application through a low-energy method in which phase transitions play a major role (*Solans & Solé, 2012*), this approach may also be considered advantageous.

Both prototypes were previously diluted with deionized water to avoid multiple scattering as follows: polysorbate 85 (HLB 11) $-1:60$ and polysorbate 80 + sorbitan monooleate (HLB 13) $-1:10$. Table 1 presents the results of hydrodynamic size expressed in terms of intensity and polydispersity index (mean $\pm$standard deviation). The zeta potential of the nanoemulsion was also measured, with average values as follows: $-32,84$ mV for polysorbate 85 (HLB 11) and $-13,28$ mV for polysorbate 80 + sorbitan monooleate (HLB 13).

A smaller hydrodynamic size was observed for the nanoemulsion prepared with polysorbate 80 + sorbitan monooleate (HLB 13). It also presented lower hydrodynamic size variation and no statistical difference ($p > 0.05$) when compared to the nanoemulsion prepared with P85 (HLB 11) ($p < 0.01$). Regarding the polydispersity index, no statistically significant difference was observed in the nanoemulsion prepared with P85 (HLB 11)

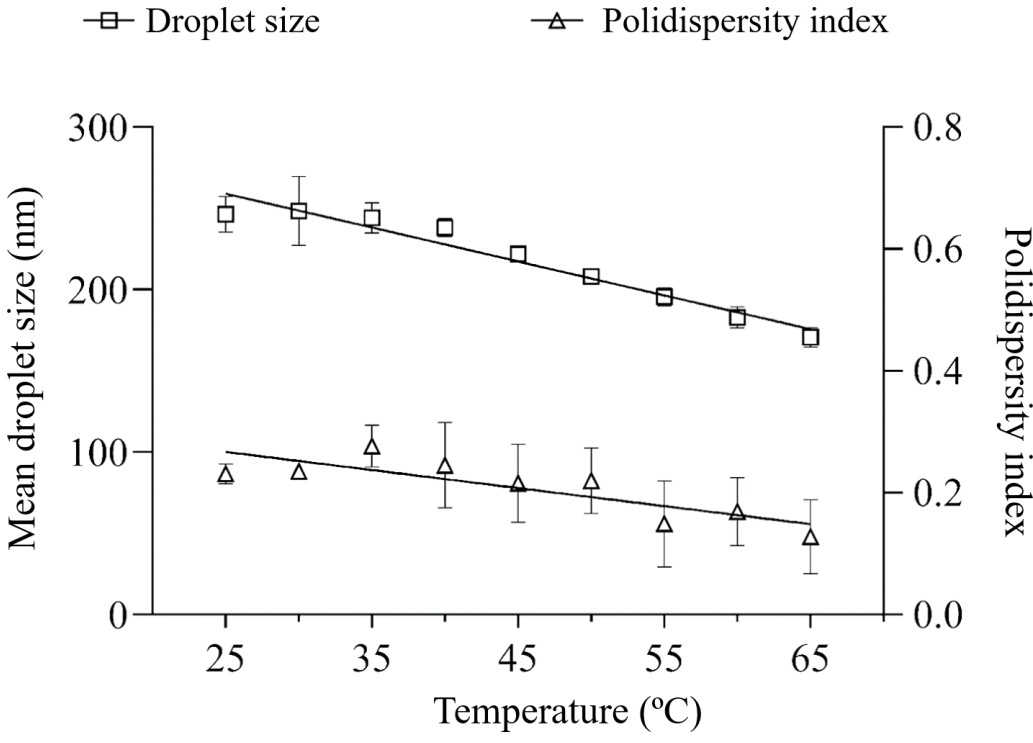

**Figure 2** **Hydrodynamic size distribution of the nanoemulsions measured after linear ramp of temperature increase.** The nanoemulsion was prepared at 9:1 SOR with polysorbate 85 (HLB 11).

($p > 0.05$), while a significant difference was observed in the nanoemulsion prepared with polysorbate 80 + sorbitan monooleate HLB 13) on day 7 (D0 × D7, $p > 0.05$; D1 × D7, $p > 0.01$). However, in the latter, the polydispersity index decreased to approximately 0.2. The polydispersity index is associated with the homogeneity of nanoemulsion populations, and low values are associated with monomodal distribution.

Polysorbate 85 has a chemical structure with three tails of 18 carbon atoms but only one head of ethylene oxide moiety (total = 20). Polysorbate 80 has a tail with 18 carbon atoms and three hydrophilic moieties of ethylene oxide (total = 20). The latter characteristic is responsible for a higher capacity of attracting water molecules that can cover the oil nanoemulsions (*Khatri & Shao, 2018*). Adsorption of chemical entities can modulate the zeta potential, therefore providing optimal repulsive forces between droplets. This was previously observed for complex amazon oils, in which the conjugated base of acid compounds would be adsorbed in the thin layer around droplets. An additional mechanism for better performance of polysorbate 80/sorbitan monooleate nanoemulsion at HLB 13 may be an optimal amount of sorbitan monooleate, which has a low HLB value (4.3) and therefore being a lipophilic surfactant, that can inhibit the release of chemical compounds from small to larger droplets through external phase, thus avoiding Ostwald ripening.

The influence of thermal stress on the nanoemulsions was performed subjecting them to a linear ramp of temperature increase. A greater tendency for mean droplet size variation

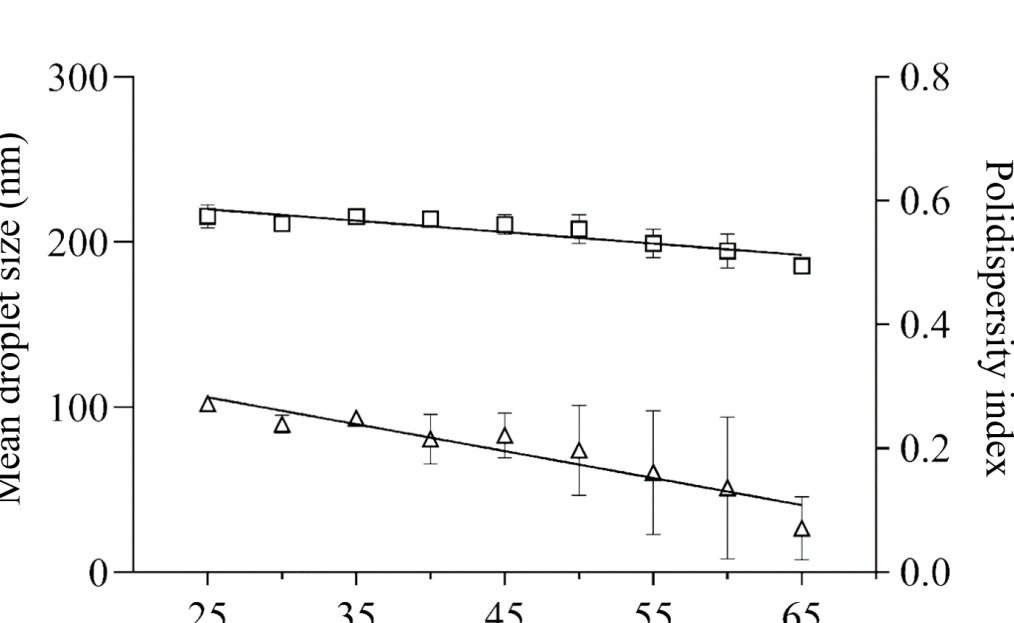

**Figure 3** Droplet size distribution after linear ramp of temperature increase of the nanoemulsion prepared at SOR 9:1 with polysorbate 80 + sorbitan monooleate (HLB 13).

was observed for HLB 11 (30.8%), compared to HLB 13 (13.9%) and no variation ($p > 0.05$) in the polydispersity index was observed (Figs. 2 and 3).

Ostwald ripening (OR) is a primary breakdown mechanism of nanoemulsion destabilization, where components migrate from smaller to larger particles. The slope of linear regression of nanoemulsion size cubed *versus* time provides an estimate of OR (*Sadeghian et al., 2023*). In the present study, no main signs of nanoemulsion growth associated to size distribution graphs were observed. One would expect that even with nanoemulsion reduction behavior, a lower slope would be observed for a more stable nanoemulsion. In this context, the equations of nanoemulsion size *versus* temperature corroborate the better physical performance of the nanoemulsion with polysorbate 80 and sorbitan monooleate ($Y = -0{,}6920{*}X + 237{,}1$) compared to the nanoemulsion prepared solely with polysorbate 85 ($Y = -2{,}087{*}X + 311{,}2$).

Previous studies highlight that nanostructures have an economic impact close to trillion dollars (*Faria-Tischer & Tischer, 2012*) Analysis of the market regarding cosmetics giants shows an increase of 7.14% annually (*Nagpal, 2023*). Regarding products encapsulating herbal-derivatives, it is worth mentioning that formulation with adjuvants that are accepted by the Food and Drug Administration (FDA) is an advantage (*Jamir et al., 2024*). Therefore, this paper highlights the potential of a nanoemulsion designed and analyzed with açaí oil, an important amazon raw material, with food-grade non-ionic surfactants, indicating

that a low energy, low cost, solvent-free and non-heating method is capable of obtaining suitable nanoemulsions for further cosmetic utilization.

## CONCLUSIONS

In conclusion, in this study we successfully demonstrated the formulation and characterization of açaí-based nanoemulsions using a low-energy method. By evaluating various surfactants and their ratios, it was found that the combination of polysorbate 80/sorbitan monooleate at HLB 13 at an SOR of 9:1 was the most effective in creating stable nanoemulsions with small droplet sizes and minimal size variation over time. This was especially compared to a system prepared solely with polysorbate 85, highlighting the blend capability of inhibiting Ostwald ripening, probably due the ability of sorbitan monooleate generating strengthening the film. The utilization of a solvent-free/non-heating method is in accordance with ecofriendly concepts and also reduces probable degradation of the product, in addition to reduced costs for industrial purposes. The findings contribute to the knowledge on natural product-based nanoemulsions, emphasizing the importance of surfactant selection in achieving desirable physicochemical properties. Future studies should explore the long-term stability of these nanoemulsions and their efficacy in cosmetic applications, potentially leading to innovative, eco-friendly skincare products that capitalize on the rich biodiversity of the Amazon region.

### Funding

This work was supported by the Brazilian agencies Conselho Nacional de Desenvolvimento Científico e Tecnológico (CNPq), Coordenação de Aperfeiçoamento de Pessoal de Nível Superior (CAPES), and Fundação Carlos Chagas Filho de Amparo à Pesquisa do Estado do Rio de Janeiro (FAPERJ). The Multiusuário de Caracterização de Materiais (LAMATE) at Universidade Federal Fluminense also supported this work. The funders had no role in study design, data collection and analysis, decision to publish, or preparation of the manuscript.

### Grant Disclosures

The following grant information was disclosed by the authors:
The Brazilian agencies Conselho Nacional de Desenvolvimento Científico e Tecnológico (CNPq).
Coordenação de Aperfeiçoamento de Pessoal de Nível Superior (CAPES).
Fundação Carlos Chagas Filho de Amparo à Pesquisa do Estado do Rio de Janeiro (FAPERJ).
The Multiusuário de Caracterização de Materiais (LAMATE) at Universidade Federal Fluminense.

### Competing Interests

The authors declare there are no competing interests.

## Author Contributions

- Mikaela Ferreira conceived and designed the experiments, performed the experiments, analyzed the data, prepared figures and/or tables, authored or reviewed drafts of the article, and approved the final draft.
- Leandro Machado Rocha analyzed the data, authored or reviewed drafts of the article, and approved the final draft.
- Rodrigo Cruz performed the experiments, analyzed the data, performed the computation work, prepared figures and/or tables, and approved the final draft.
- Francisco Paiva Machado performed the experiments, analyzed the data, performed the computation work, prepared figures and/or tables, authored or reviewed drafts of the article, and approved the final draft.
- Celia Machado Ronconi analyzed the data, authored or reviewed drafts of the article, and approved the final draft.
- Caio Fernandes conceived and designed the experiments, performed the experiments, analyzed the data, authored or reviewed drafts of the article, and approved the final draft.

## Data Availability

The raw measurements are available in the Supplementary File.

## Supplemental Information

Supplemental information for this article can be found online at http://dx.doi.org/10.7717/peerj-ochem.13#supplemental-information.

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
