# Peer review of "Understanding colloidal behavior in açaí (*Euterpe oleraceae*)-based nanoemulsions"

_PeerJ Organic Chemistry, doi:10.7717/peerj-ochem.13_

## Round 0.1 · original submission · Major Revisions

Dear Author,

The manuscript is well-written but requires significant revisions based on the reviewers’ feedback. Key areas to address include:

Basic Reporting: The abstract should be revised to eliminate redundancies. Figures and tables need clearer labels and detailed captions. Statistical significance should be mentioned where relevant.

Experimental Design: Clarify the novelty of the work. Justify the choice of surfactants over others and include a control sample (e.g., açaí oil without surfactants). Provide a clear, quantitative explanation for the 9:1 surfactant-to-oil ratio.

Methodology: Provide a stepwise description of nanoemulsion synthesis and include details on sample preparation, storage, and measurement repetitions for reproducibility.

Discussion and Validity: Improve the discussion by comparing findings with existing literature, explaining molecular interactions responsible for stability, and discussing potential industrial/commercial implications. Acknowledge study limitations.

Action Required: Revise the manuscript addressing all comments suggested by reviewers. The revised version will be reconsidered for publication.

**Language Note:** PeerJ staff have identified that the English language needs to be improved. When you prepare your next revision, please either (i) have a colleague who is proficient in English and familiar with the subject matter review your manuscript, or (ii) contact a professional editing service to review your manuscript. PeerJ can provide language editing services - you can contact us at [email protected] for pricing (be sure to provide your manuscript number and title). – PeerJ Staff

·

Basic reporting

The manuscript is generally well-written but has several grammatical and typographical errors. Refinement in sentence structures and proper phrasing will enhance clarity.
The introduction provides an adequate background, but more recent studies on nanoemulsions and their applications in cosmetics and pharmaceuticals should be referenced to establish novelty.
The figures and tables are informative but should be better labeled with detailed captions for clarity. Statistical significance should be explicitly mentioned where applicable.

Experimental design

The study addresses a relevant topic in nanotechnology and sustainable product development. However, the knowledge gap should be more explicitly stated in relation to previous studies.
The experimental procedures are described with moderate detail. However, additional specifics on sample preparation, storage conditions, and measurement repetitions would enhance reproducibility.

Validity of the findings

The surfactant selection criteria should be explained more clearly, why were certain surfactants chosen over others?
The results are well-structured but should include a more thorough discussion comparing findings with similar studies.
The conclusions align with the research objectives, but limitations of the study should be acknowledged to provide a balanced perspective.

Reviewer 2 ·

Basic reporting

The article “Understanding colloidal behavior in açaí (Euterpe oleraceae)-based nanoemulsions” is well written but needs some changes to be addressed, which are mentioned below:
1. In Abstract section, Line# 1 and 2: redundant comma and repetition of word “products” should be corrected.

Experimental design

2. The authors stated the hydrophilic-lipophilic balance (HLB) values of surfactants but the reason why these were preferred over the already reported compounds is not clear.
3. Study including the control or reference sample (e.g., an açaí oil emulsion without surfactants) is missing, that will support for comparing the results.

Validity of the findings

4. The authors mentioned numerous surfactant-to-oil ratios but did not elaborated how the final choice (9:1) was determined quantitatively.
5. The discussion part does not clearly elucidates the molecular interactions responsible for stability. How the surfactants stabilize the emulsions should be given in detail which would strengthen the findings.

Annotated reviews are not available for download in order to protect the identity of reviewers who chose to remain anonymous.
Cite this review as

Reviewer 3 ·

Basic reporting

The research question is clearly defined and addresses a relevant knowledge gap in nanoemulsion formulation for cosmetic applications.
Discussion of the results is poor. The findings have not been discussed with suitable reasons and not compared with literature.

Experimental design

Novelty of this work is not clear.
Experimental methods do not have sufficient details. Specifically, synthesis of nanoemulsions should be stated in stepwise manner without an ambiguity.

Validity of the findings

A brief discussion on the potential industrial or commercial implications of the nanoemulsions reported in this article may increase the impact of this article.

Cite this review as

---

## Round 0.2 · accepted · Accept

As the authors have addressed the revisions in a professional manner. The manuscript is well-structured, the language is clear, and the content is relevant to the journal’s scope. The research question is clearly identified, and the experimental approach is appropriate. The findings are valuable, and the conclusions are well-supported by the results so the manuscript is accepted for publication in its present form.

Reviewer 3 ·

Basic reporting

Authors have revised the article in a professional manner. The introduction provides a background of the work, research gap and hypothesis to address that research gap. Recent and relevant literate has been cited and critically discussed. The results are valuable and have been discussed with scientific reasons. Overall, the article is well structured, and language is easily understandable.

Experimental design

This work is related to chemistry and lies within the aims and scope of the journal. The research question of impaired stability or complex mixtures of natural products in cosmetics was identified by the authors clearly. Authors have introduced a suitable scientific approach for to address the question by using nanoemulsion. The performed investigation is sufficient for future work in this direction. Experimental part contains sufficient details to repeat this work.

Validity of the findings

Authors have successfully demonstrated the formulation and characterization of açaí-based nanoemulsions using a low-energy method. The findings contribute to the knowledge use of nanotechnology and natural products in cosmetics. All the related data has been provided by the authors, and it seems valuable. Author have compared their results with literature and comparison shows that current results are more beneficial than the previous ones. The conclusions are linked to the results and well written.

Additional comments

The revised article is suitable for publication.

Cite this review as